# Hunger neurons drive feeding through a sustained, positive reinforcement signal

Yiming Chen[1,2,3], Yen-Chu Lin[1,2], Christopher A Zimmerman[1,2,3], Rachel A Essner[1,2], Zachary A Knight[1,2,3]*

[1]Department of Physiology, University of California, San Francisco, San Francisco, United States; [2]Kavli Institute for Fundamental Neuroscience, University of California, San Francisco, San Francisco, United States; [3]Neuroscience Graduate Program, University of California, San Francisco, San Francisco, United states

**Abstract** The neural mechanisms underlying hunger are poorly understood. AgRP neurons are activated by energy deficit and promote voracious food consumption, suggesting these cells may supply the fundamental hunger drive that motivates feeding. However recent in vivo recording experiments revealed that AgRP neurons are inhibited within seconds by the sensory detection of food, raising the question of how these cells can promote feeding at all. Here we resolve this paradox by showing that brief optogenetic stimulation of AgRP neurons before food availability promotes intense appetitive and consummatory behaviors that persist for tens of minutes in the absence of continued AgRP neuron activation. We show that these sustained behavioral responses are mediated by a long-lasting potentiation of the rewarding properties of food and that AgRP neuron activity is positively reinforcing. These findings reveal that hunger neurons drive feeding by transmitting a positive valence signal that triggers a stable transition between behavioral states.

*For correspondence: zachary.knight@ucsf.edu

**Competing interests:** The authors declare that no competing interests exist.

## Introduction

Food deprivation motivates animals to find and consume food. This implies that the brain can transform nutritional signals into the desire to eat, but how this transformation is performed remains unclear.

Agouti-related protein (AgRP) neurons within the arcuate nucleus (ARC) of the hypothalamus are a molecularly-defined cell type that is particularly important for the control of feeding. AgRP neurons are regulated by hormones that report on the nutritional state of the body (*Cowley et al., 2001*, *2003*; *Gao and Horvath, 2007*; *Nakazato et al., 2001*; *Pinto et al., 2004*) and their activity is strongly increased by food deprivation (*Hahn et al., 1998*; *Mandelblat-Cerf et al., 2015*). Optogenetic or chemogenetic stimulation of AgRP neurons promotes intense food consumption as well as appetitive behaviors that lead to food discovery (*Aponte et al., 2011*; *Krashes et al., 2011*), whereas inhibition of these neurons leads to aphagia (*Gropp et al., 2005*; *Krashes et al., 2011*; *Luquet et al., 2005*). Thus AgRP neurons are poised to connect nutritional signals with the motivational processes that govern feeding.

Traditionally, AgRP neurons were thought to be regulated primarily by nutritional cues that circulate in the blood (*Gao and Horvath, 2007*; *Luo, 2015*). According to this model, AgRP neurons are activated by gradual changes in the concentrations of hormones such as leptin and ghrelin that develop during food deprivation. This generates a "hunger drive" that motivates animals to find and consume food and persists until food consumption restores these hormones to their previous level, thereby inhibiting AgRP neurons and quelling the desire to eat.

Recently, this textbook model was challenged by experiments that recorded for the first time the activity of AgRP neurons in awake, behaving mice (*Betley et al., 2015*; *Chen et al., 2015*;

*Mandelblat-Cerf et al., 2015*). These experiments unexpectedly revealed that AgRP neurons are inhibited within seconds by the mere sight and smell of food, or by conditioned cues that predict food availability. These responses were much too fast to be mediated by a hormonal signal, implying that they arise from changes in neural input. Paradoxically, this rapid inhibition was often complete before a single bite of food could be consumed, such that AgRP neuron activity was greatly reduced prior to the onset of feeding. This observation raises the question of how AgRP neurons are able to drive feeding at all.

Several hypotheses have been advanced to explain these counterintuitive findings (*Chen and Knight, 2016*; *Seeley and Berridge, 2015*). An important unresolved question regards when AgRP neuron activity must occur in order to influence feeding behavior. While it has long been assumed that AgRP neurons promote feeding primarily through firing that occurs during the act of food intake (*Aponte et al., 2011*), an alternative possibility is that AgRP neuron activity before food obtainment could be sufficient to elicit the voracious feeding that occurs later (*Chen and Knight, 2016*). If such a mechanism were operational, then it would explain how AgRP neurons could promote food intake despite being silenced at the beginning of a meal by sensory cues.

Here we investigate this question by using optogenetics to stimulate AgRP neurons selectively before food availability, thereby "replaying" the natural regulation of these cells that occurs during fasting and refeeding. We find that this preparatory photostimulation is sufficient to elicit voracious food consumption and vigorous operant responding for food in well-fed animals. These sustained behavioral effects develop rapidly, persist for tens of minutes, and can be triggered by stimulation of several distinct anatomic pathways. We show that these long-lasting behavioral changes are mediated by a motivational switch that magnifies the positively rewarding properties of food, and furthermore that AgRP neuron activity is positively reinforcing. These findings reconcile the function of AgRP neurons with their paradoxical natural dynamics, and in doing so reveal the motivational mechanism by which these cells drive food consumption.

## Results

### AgRP neurons transmit a sustained hunger signal

To test the hypothesis that AgRP neurons drive feeding through a sustained mechanism, we used optogenetics to manipulate AgRP neuron activity before food presentation and then measured the effect on subsequent feeding behavior (*Figure 1B*). Ad libitum fed mice expressing channelrhodopsin in AgRP neurons (AgRP-ChR2; *Figure 1C*) were acclimated to a behavioral chamber early in the light phase, a time when mice ordinarily eat little, and photostimulated for one hour in the absence of food. Photostimulation was then terminated and food was made available (*Figure 1B*). Strikingly, we found that this preparatory photostimulation triggered intense feeding upon subsequent food presentation (*Figure 1D–G*). This voracious feeding approached the level of food consumption observed following an overnight fast (*Figure 1E*); it did not require learning, as it was observed in the first trial of every mouse (*Figure 1—figure supplement 1*); and it was absent from control mice that lacked ChR2 expression (*Figure 1—figure supplement 1*). Thus stimulation of AgRP neurons in the absence of food is sufficient to elicit intense food consumption at a later time when food is made available. Importantly, this observation provides an explanation for how AgRP neurons can promote feeding despite being inhibited at a meal's outset by the sensory detection of food (*Figure 1A*).

We investigated the properties of this sustained feeding response. Prestimulation of AgRP neurons for as little one minute was sufficient to increase food intake above the baseline level of unstimulated mice (0.34 ± 0.03 vs 0.14 ± 0.03 g, p<0.01), indicating that the response begins to develop rapidly. Increasing the duration of prestimulation progressively increased the amount of food consumed, reaching a plateau at approximately 30 min (*Figure 1F*). This relationship between prestimulation duration and subsequent food intake displayed first order association kinetics (*Figure 1G*; $R^2$ = 0.96), suggesting that AgRP neuron activity transmits a saturable signal that "builds up" in a downstream circuit element. Consistent with this model, the sustained effects of AgRP neuron activity did not require a specific sequence of light pulses (*Figure 1—figure supplement 1*), although intermittent, high frequency stimulation (20 Hz) was slightly more effective than tonic lower frequency stimulation (10 Hz) at eliciting feeding when the total number of light pulses was held

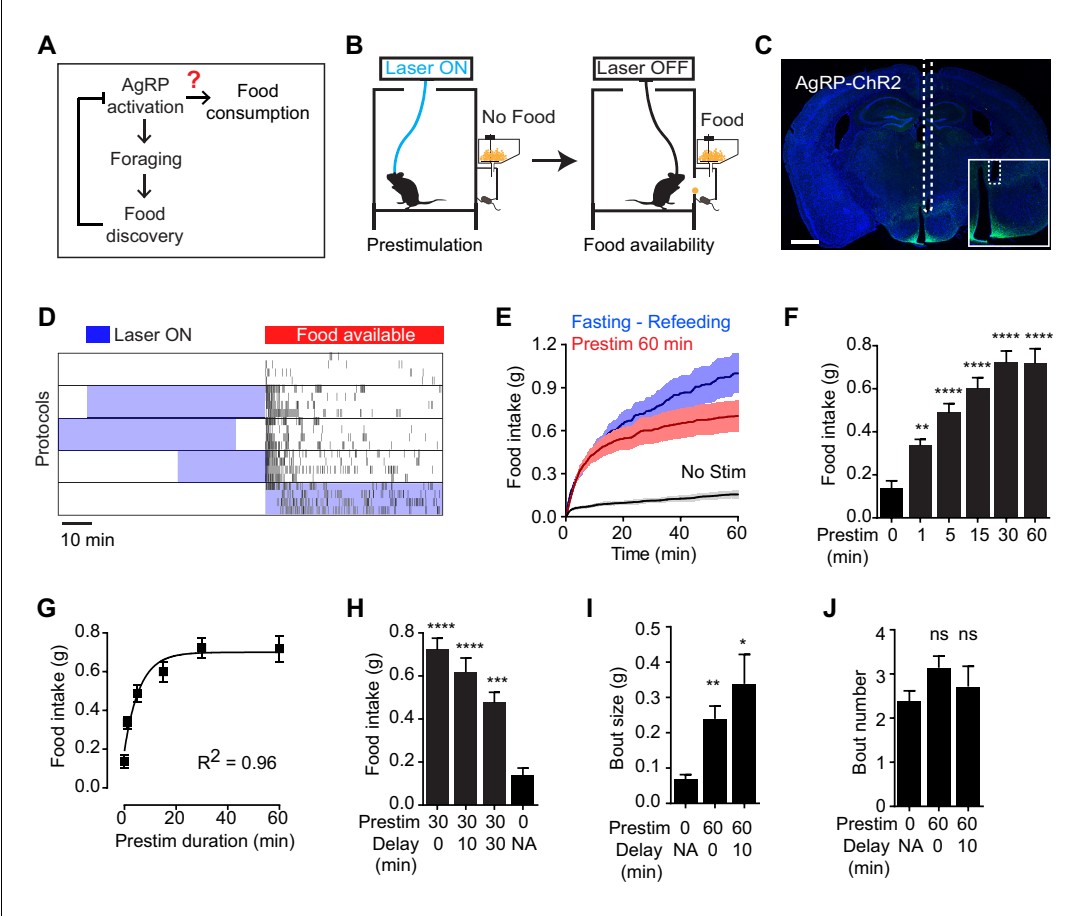

**Figure 1.** Prestimulation of AgRP neurons promotes sustained consummatory behavior. (A) Current model of feeding control by AgRP neurons illustrating the disconnect between the natural dynamics and orexigenic function of these cells. (B) Schematic of the prestimulation experiment. (C) Expression of ChR2-eYFP in AgRP neurons and optical fiber placement above arcuate nucleus. (D–J) Prestimulating ARC$^{AgRP}$ neurons evokes food consumption in fed mice. (D) Raster plots showing temporal relationship between food pellet consumption events (black vertical bars) and opto-stimulation patterns (blue boxes). (E) Plots of cumulative food intake by mice after 0 min prestim (black n = 7), 60 min prestim (red n = 7) and overnight fasting (blue n = 6). Filled areas indicate S.E.M. (F) Food intake evoked by prestimulation with varied duration (n = 8). (G) First order association between average total food intake and prestimulation durations (Equation: Y = Y0 + (Plateau-Y0)*(1-exp(-K*x))). (H) Food intake evoked by protocols with different duration of delay between prestimulation and food availability (n = 8). (I) Bout size analysis of different prestimulation protocols (n = 7). (J) Bout number analysis of prestimulation protocols (n = 7). Asterisks on top of bar plots indicate significance levels compared to no stimulation control and asterisks on top of brackets indicate significance levels for comparisons with the respective protocols, using one-way-ANOVA with Holm-Sidak's correction for multiple comparisons (****p≤0.0001, ***0.0001<p≤0.001, **0.001<p≤0.01, *0.01<p≤0.05, ns p>0.05).

The following figure supplement is available for figure 1:

**Figure supplement 1.** Prestimulation of AgRP neurons primes feeding.

constant. These sustained effects were long-lasting, as the introduction of a delay of 30 min between the offset of AgRP neuron stimulation and onset of food availability only modestly decreased subsequent food consumption (*Figure 1H*). Analysis of the microstructure of feeding revealed that these long-lasting effects were manifest primarily as an increase in bout size, rather than bout number (*Figure 1I,J*). Thus AgRP neuron activity transmits a hunger signal that accumulates in a downstream circuit element on a timescale of approximately 30 min, resulting in a sustained potentiation of feeding that persists even after AgRP neurons have been silenced.

# Prestimulation of thirst neurons does not have a sustained effect on drinking

We wondered whether this unusual persistent mechanism utilized by AgRP neurons to drive feeding is a general feature of neurons that control ingestive behavior. To test this, we examined thirst-promoting neurons in the subfornical organ that express Nos1 (SFO$^{Nos1}$ neurons). SFO$^{Nos1}$ neurons are activated by circulating signals of fluid balance, and their artificial stimulation is sufficient to drive voracious drinking even in water sated animals (*Betley et al., 2015*; *Oka et al., 2015*; *Zimmerman et al., 2016*). We delivered ChR2 to SFO$^{Nos1}$ neurons by stereotaxic injection of a Cre dependent AAV into the SFO of Nos1-IRES-Cre mice (*Figure 2A*), and then measured the effects of optogenetic stimulation of these cells on water consumption (*Figure 2B*). As previously reported, photostimulation of SFO$^{Nos1}$ neurons resulted in rapid and intense drinking (508 ± 68 licks for stimulated animals versus 3 ± 1 licks for controls, $p<0.001$ *Figure 2C,D*). However, unlike AgRP neurons, prestimulation of SFO$^{Nos1}$ neurons for one hour prior to water access had no effect on subsequent water intake (*Figure 2C,D*). We confirmed that this was not due to a technical problem associated with chronic photostimulation of these cells, because 30 min of prestimulation did not impair the ability of subsequent co-stimulation in the presence of water to drive drinking (*Figure 2D*, column 3). Thus the persistent mechanism utilized by AgRP neurons to promote feeding does not generalize to related ingestive circuits.

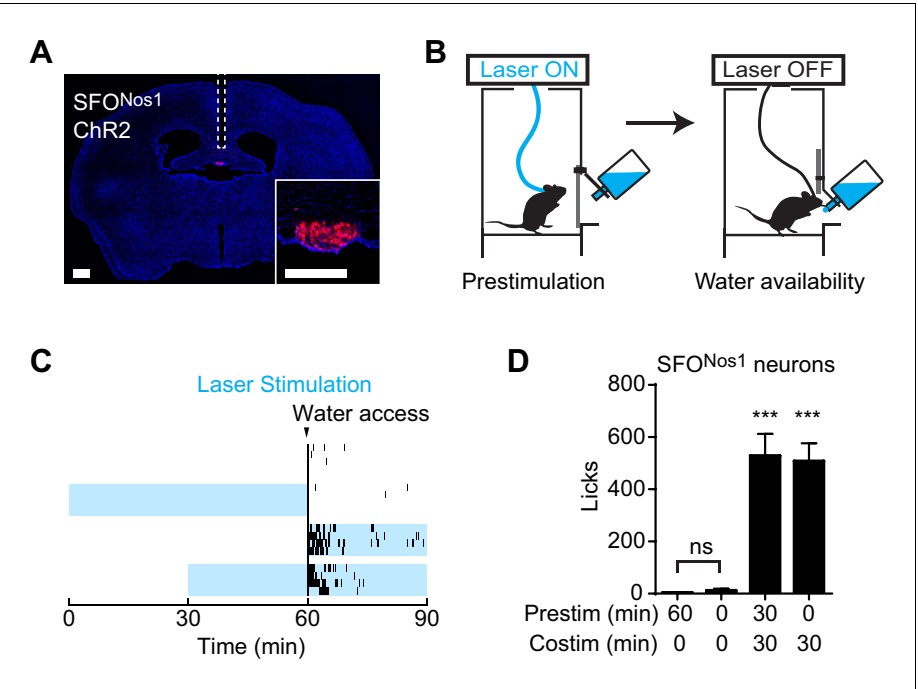

**Figure 2.** Prestimulation of SFO$^{Nos1}$ neurons does not prime drinking behavior. (**A**) Expression of ChR2-mCherry in SFO$^{Nos1}$ neurons and optical fiber placement above SFO. (**B**) Schematic of the prestimulation experiment. (**C–D**) Drinking evoked by different protocols stimulating SFO$^{Nos1}$ neurons (n = 4). (**C**) Raster plots showing temporal relationship between licking (black vertical lines) and opto-stimulation pattern (blue boxes). (**D**) Comparison of total licking events. Co-stimulation data are a reanalysis of experiments described in (*Zimmerman et al., 2016*). Asterisks on top of bar plots indicate significance levels compared to no stimulation control and asterisks on top of brackets indicate significance levels for comparisons with the respective protocols, using one-way-ANOVA with Holm-Sidak's correction for multiple comparisons (****$p\leq0.0001$, ***$0.0001<p\leq0.001$, **$0.001<p\leq0.01$, *$0.01<p\leq0.05$, ns $p>0.05$).

## Prestimulation of AgRP neurons results in sustained motivation to work for food

AgRP neurons promote not only food intake but also appetitive behaviors that lead to food obtainment. For example, optogenetic or chemogenetic stimulation of AgRP neurons motivates animals to perform instrumental responses such as lever pressing in order to obtain a food reward (*Atasoy et al., 2012*; *Krashes et al., 2011*). It has been hypothesized that the rapid inhibition of AgRP neurons by the sensory detection of food may serve as a signal that inhibits these appetitive behaviors, thereby enabling the transition from foraging to feeding (*Chen and Knight, 2016*). However, this possibility has never been directly tested.

To investigate this question, we tested whether prestimulation of AgRP neurons would alter animals' subsequent motivation to work for food. We trained AgRP-ChR2 mice to lever press for food pellets and then tested them in a progressive ratio 3 (PR3) reinforcement schedule (*Figure 3A*), in which an increasing number of lever presses are required to obtain each successive food reward (*Hodos, 1961*). In the absence of prior photostimulation, AgRP-ChR2 mice engaged in a low level of operant responding (*Figure 3C,D*), consistent with the fact that ad libitum fed mice have little motivation to work for food. In contrast, prestimulation of AgRP neurons for one hour caused animals to engage in vigorous pressing when the levers were subsequently made available (*Figure 3B,C*). This operant responding was specifically directed toward the food reward, because animals pressed the active lever much more frequently than the inactive lever ($223 \pm 29$ for active lever vs. $28 \pm 6$ inactive, $p<0.001$; *Figure 3B,C*). This response was also specific to AgRP neuron activation, because it was absent from sham stimulated mice that lacked ChR2 expression (*Figure 3D*). Thus prestimulation of AgRP neurons generates long-lasting motivation to work for food that persists even in the absence of continued AgRP neuron activity. This indicates that both appetitive and consummatory behaviors can be driven by a sustained signal from AgRP neurons.

## AgRP neuron projections to the PVH, BNST and LHA are individually sufficient to generate persistent hunger

We next investigated the neural pathway that underlies these sustained behavioral effects. AgRP neurons project to several downstream targets in a primarily one-to-one configuration (*Betley et al., 2013*). Among these, the paraventricular hypothalamus (PVH), bed nucleus of the stria terminalis (BNST), and lateral hypothalamic area (LHA) are particularly strongly innervated by AgRP neuron axons (*Broberger et al., 1998*). Optogenetic stimulation of AgRP neuron terminals in each of these

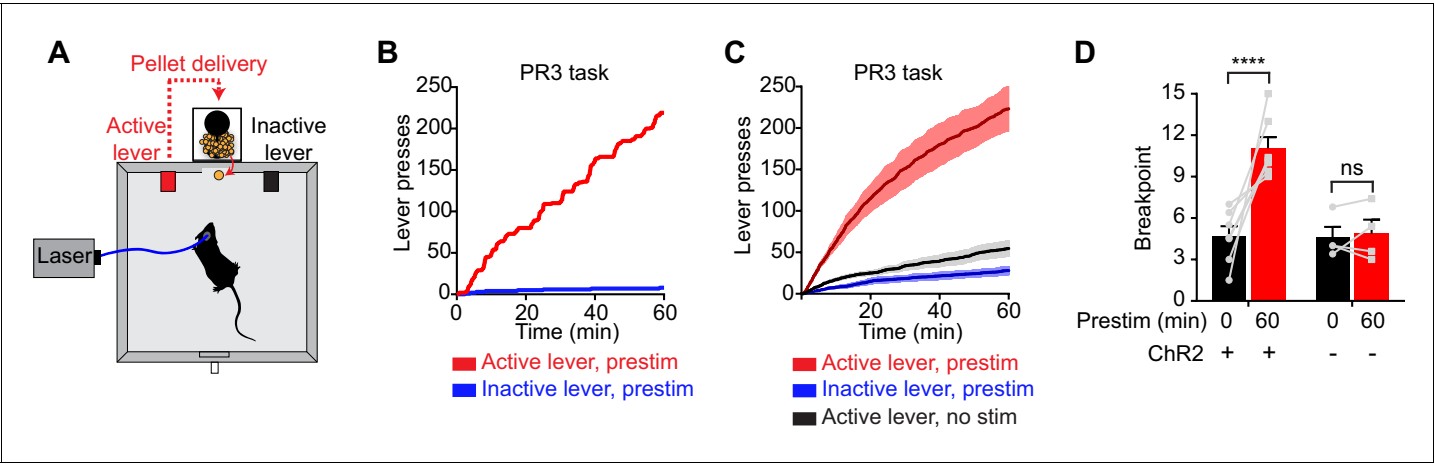

**Figure 3.** Prestimulation of AgRP neurons promotes sustained appetitive behavior. (**A**) Progressive ratio 3 lever press task. (**B–D**) Lever presses evoked by prestimulating AgRP neurons in progressive ratio 3 tasks (n = 7). (**B**) Plots of cumulative lever presses by a representative mouse after 60 min prestimulation. (**C**) Plots of average cumulative lever presses from trials with or without prestimulation. Filled areas indicate S.E.M. (**D**) Breakpoint in a lever pressing assay for ChR2+ and ChR2- mice. Asterisks on top of brackets indicate significance levels for comparisons with the respective protocols, using one-way-ANOVA with Holm-Sidak's correction for multiple comparisons (****$p \leq 0.0001$, ***$0.0001 < p \leq 0.001$, **$0.001 < p \leq 0.01$, *$0.01 < p \leq 0.05$, ns $p > 0.05$).

three areas during the act of feeding can drive voracious food consumption (*Betley et al., 2013*), but it remains unknown whether these same projections support feeding under more physiologic stimulation conditions, in which AgRP neurons are highly active only before food availability.

To test this, we implanted optical fibers above the PVH, BNST, or LHA of AgRP-ChR2 mice and then measured food intake following one hour of preparatory photostimulation (*Figure 4*). We found that prestimulation of AgRP neuron axons in all of these regions elicited robust food intake compared to non-stimulated controls (PVH: 0.55 ± 0.05 g vs. 0.11 ± 0.02 g, p<0.0001; BNST: 0.38 ± 0.06 g vs. 0.14 ± 0.03 g, p<0.01; LHA: 0.60 ± 0.01 g vs. 0.21 ± 0.05 g, p<0.001 *Figure 4* and *Figure 4—figure supplement 1*). Quantitative analysis of the relationship between prestimulation duration and food intake for ARC → PVH projections revealed that this sustained response built up progressively over time (*Figure 4—figure supplement 1*), with kinetics similar to those observed for prestimulation of the soma (*Figure 1F*). These sustained effects were unaffected by the introduction of a 10 min delay between laser offset and the onset of food availability, indicating that they persist in the absence of ongoing feeding behavior (*Figure 4* and *Figure 4—figure supplement 1*). To investigate the role of these pathways in appetitive behaviors, we prestimulated each projection for one hour and then measured lever pressing in a progressive ratio assay. All three projections supported vigorous and specific lever pressing (*Figure 4* and *Figure 4—figure supplement 1*) to an extent that was comparable to the effect of stimulating all AgRP neurons in the ARC (*Figure 4H*). Thus, at least three different projections of AgRP neurons are individually sufficient to elicit the sustained feeding behavior that arises from AgRP neuron activity.

## Prestimulation of AgRP neurons conditions appetite and flavor preference

The fact that AgRP neurons can drive feeding through a sustained mechanism implies that the underlying motivational processes must also be long-lasting. However the nature of the motivational signals that persist after AgRP neurons have been silenced is unknown. One important mechanism by which food deprivation motivates feeding is by enhancing the positively rewarding properties of food, such as its palatability (*Berridge, 2004*, *2009*; *Cabanac, 1971*; *Fulton, 2010*; *Lockie and Andrews, 2013*; *Rolls et al., 1980*). We therefore considered the possibility that AgRP neuron activity might promote long-lasting potentiation of food's incentive value.

To test this, we investigated whether AgRP neuron prestimulation could condition appetite for specific foods. AgRP-ChR2 mice were acclimated to a feeding chamber that delivered pellets that had a similar energy density to their home cage chow, but had a distinct size, shape, and texture (*Figure 5A*; see Methods for additional information). We then tested mice in this chamber for pellet consumption during a one hour test period on eight consecutive days (*Figure 5B*). The trial was designed so that, on days 3, 5, and 7, the test period was immediately preceded by one hour of AgRP neuron prestimulation (*Figure 5B*, blue), whereas on the intervening days (days 1, 2, 4, 6, and 8), there was mock stimulation. Of note, mice had ad libitum access to chow in their home cage, and all animals were laser naïve at the beginning of the trial, meaning that day 3 was the first time they were exposed to photostimulation.

We found that mice consumed relatively few test pellets at baseline (trials 1 and 2; preconditioning), consistent with the fact that fed mice eat little during the light phase (*Figure 5B,C*, red). In trial 3, mice were prestimulated for one hour, and, as described above, this resulted in voracious pellet consumption (*Figure 5B* red). Strikingly, this pellet consumption remained strongly elevated in subsequent trials 4 and 6, even though these trials were not preceded by AgRP neuron stimulation (*Figure 5B,C* red; post-conditioning). This conditioned appetite was specific to the test conditions associated with AgRP neuron prestimulation, because it was not prevented by ad libitum access to chow in the home cage (*Figure 5A*) and was completely absent from sham stimulated control mice (*Figure 5B*, gray). This indicates that a single trial of AgRP neuron prestimulation can generate conditioned appetite for subsequently presented food.

We hypothesized that this conditioned appetite might reflect attribution of incentive value to the specific sensory properties of the test pellets (e.g. their taste or texture), caused by the fact that exposure to these pellets was experimentally paired with AgRP neuron prestimulation. This learned incentive value would then motivate the mice to eat those pellets in future trials even when not food deprived. A prediction of this model is that this specific appetite should undergo extinction if the test pellets are simply provided to the mice in their home cage, so that the pellets become

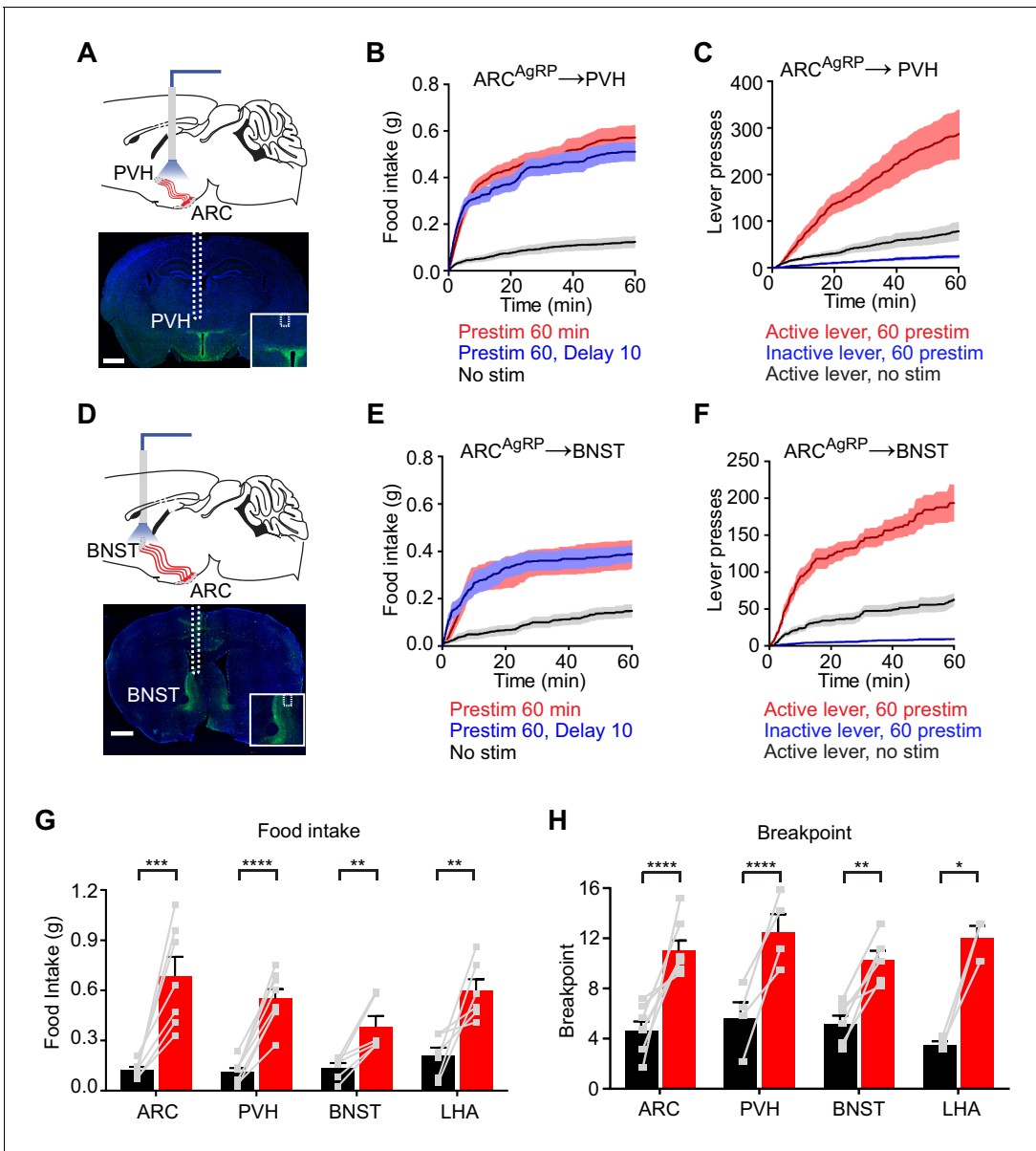

**Figure 4.** Projections of AgRP neurons to PVH, BNST or LHA are sufficient to prime feeding behavior. (**A**) Optical fiber placement above AgRP[ARC→PVH]. (**B–C**) Plots of cumulative food intake (**B**) and lever presses (**C**) evoked by prestimulating AgRP[ARC→PVH] axonal terminals. Filled areas indicate S.E.M. (**D**) Optical fiber placement above AgRP[ARC→BNST]. (**E–F**) Plots of cumulative food intake (**E**) and lever presses (**F**) evoked by prestimulating AgRP[ARC→BNST] axonal terminals. Filled areas indicate S.E.M. (**G**) 60 min food intake evoked by 60 min prestimulation of AgRP[ARC], AgRP[ARC→PVH], AgRP[ARC→BNST] and AgRP[ARC→LHA] (red) and corresponding nostim controls (black). (**H**) Breakpoint in 60 min progressive ratio 3 task reached by animals with 60 min prestimulation of AgRP[ARC], AgRP[ARC→PVH], AgRP[ARC→BNST] and AgRP[ARC→LHA] (red) and corresponding nostim controls (black). Asterisks on top of brackets indicate significance levels for comparisons with the respective protocols, using one-way-ANOVA with Holm-Sidak's correction for multiple comparisons (****$p \leq 0.0001$, ***$0.0001 < p \leq 0.001$, **$0.001 < p \leq 0.01$, *$0.01 < p \leq 0.05$, ns $p > 0.05$). ARC food intake n = 8; ARC PR3 n = 7; PVH food intake n = 8; PVH PR3 n = 4; BNST food intake n = 6; BNST PR3 n = 6; LHA food intake n = 6; LHA PR3 n = 3.

The following figure supplement is available for figure 4:

**Figure supplement 1.** Prestimulation of specific AgRP neuron projections promotes feeding.

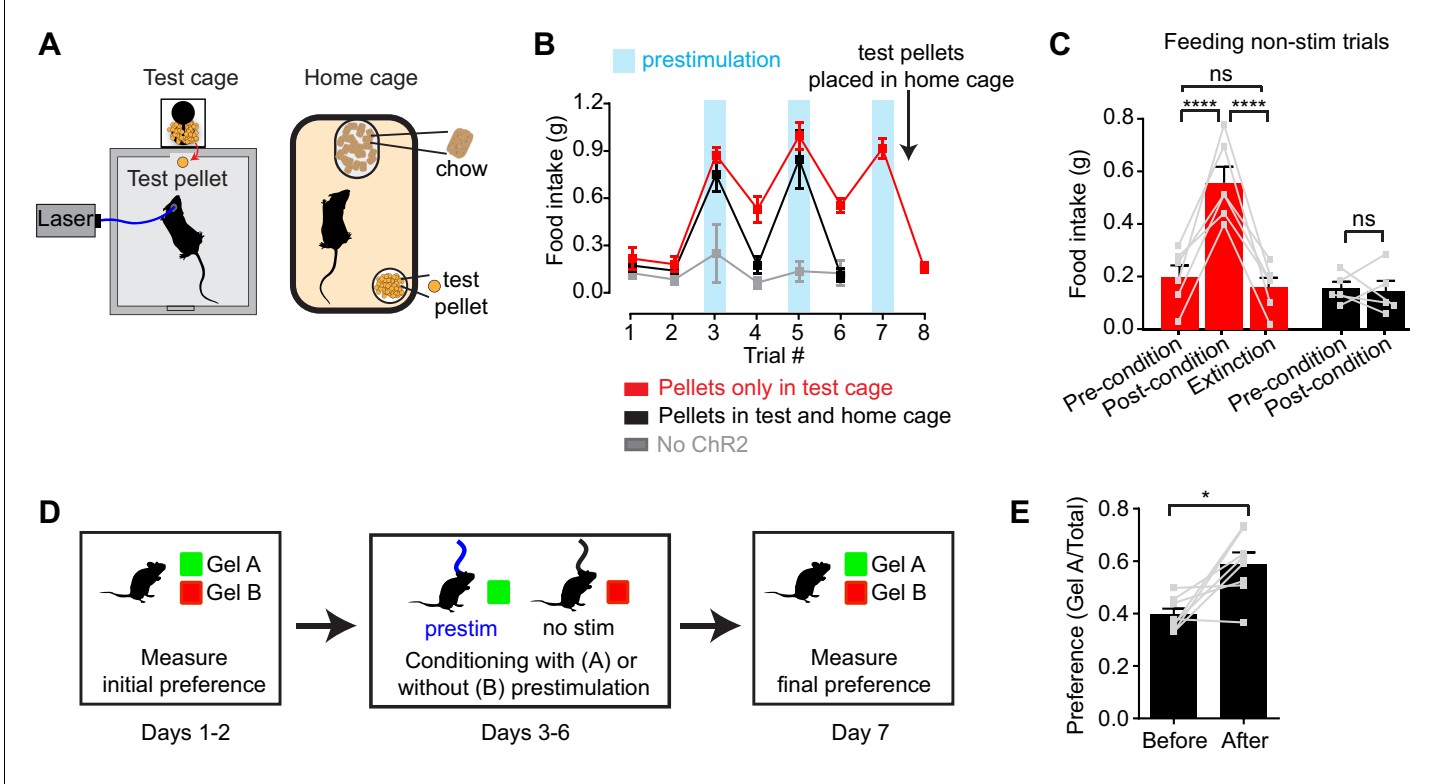

**Figure 5.** Prestimulation of AgRP neurons conditions appetite and flavor preference. (**A**) Schematic of conditioned appetite assay. Test pellets and home cage chow are similar in energy density but different in shape, size, and texture. Test pellets were either included in home cage or not, as indicated. (**B–C**) Average 60 min food intake of conditioned appetitive experiments. (**B**) Food intake of AgRP-ChR2 mice without access to test pellets in homecage (red n = 6) and with access to test pellets in homecage (black n = 5), and WT mice without access to test pellets in homecage (grey n = 3) through consecutive trials. Blue boxes indicate trials with 60 min prestimulation (trials 3,5,7), whereas in white trials animals were subjected to mock stimulation (trials 1,2,4,6,8). (**C**) Comparison among pre-conditioning, post-conditioning and extinction trials of AgRP-ChR2 mice with (black n = 5) or without (red n = 6) access to test pellets in homecage. Trial 1 and 2 are considered pre-conditioning, trial 4 and 6 are considered post-conditioning and trial 8 is considered extinction. (**D**) Conditioned flavor preference experiment. (**E**) Change of preference to conditioned flavor before and after 4 repeats of prestimulation conditioning assay (n = 8). Asterisks on top of brackets indicate significance levels for comparisons with the respective protocols, using one-way-ANOVA with Holm-Sidak's correction for multiple comparisons (****p≤0.0001, ***0.0001<p≤0.001, **0.001<p≤0.01, *0.01<p≤0.05, ns p>0.05).

dissociated from AgRP neuron prestimulation. This was indeed the case: providing the mice with overnight access to the test pellets abolished the conditioned appetite in the next trial (*Figure 5B,C* red). To test this a different way, we prepared a second cohort of laser naïve mice that were given ad libitum access to the test pellets in their home cage from the beginning of the trial (*Figure 5B,C* black). These animals showed no evidence of conditioned appetite when trained under otherwise identical conditions (*Figure 5B*, black, days 4 and 6). Thus AgRP neuron activity can condition appetite for specific foods that are consumed after AgRP neurons have shut off, such that these foods are later consumed in the absence of homeostatic need. This suggests that AgRP neuron prestimulation can attribute incentive value to the sensory properties of subsequently consumed food.

To probe this idea further, we examined whether animals could be trained to prefer a specific flavor by experimentally pairing that flavor with AgRP neuron prestimulation (*Figure 5D*). In a baseline trial, AgRP-ChR2 mice were given access two different flavors of non-caloric gels (strawberry and orange) and the amount of each consumed was recorded. The mice were then conditioned on four consecutive days by pairing access to the less preferred gel with 30 min of AgRP neuron prestimulation, whereas the preferred gel was paired with 30 min of mock stimulation (*Figure 5D*). The order of these conditioning sessions was randomized each day and they were separated by at least four hours. On day seven, the mice were then tested by providing simultaneous access to both gels and

measuring consumption of each. We found that this conditioning protocol robustly reversed the mice's flavor preference, such that the less preferred flavor became more preferred (*Figure 5E*). Thus, animals learn to prefer flavors that are preceded by AgRP neuron activation, consistent with the idea that AgRP neuron activity induces a long-lasting potentiation of the rewarding sensory properties of food.

## AgRP neuron stimulation is positively reinforcing

The preceding data suggest that AgRP neurons transmit a long-lasting, positive valence signal that potentiates the incentive value of food. The effect of this mechanism is to transform AgRP neuron firing before food availability into a sustained drive that can motivate feeding later. An important question is whether this positive valence mechanism is sufficiently strong to account for the dramatic instrumental responses (e.g. lever pressing, nose poking) that animals exhibit following AgRP neuron activation. Of note, a previous study reported that mice failed to perform operant responses in order to shut off AgRP neuron activity, indicating that these neurons do not motivate behavior by negative reinforcement (*Betley et al., 2015*), which we confirmed independently (*Figure 6—figure supplement 1*). However whether mice will perform these same actions in order to turn on AgRP neuron activity has never been tested.

We took laser naïve AgRP-ChR2 mice and acclimated them over three nights to behavioral chambers containing two levers, one of which triggered brief AgRP neuron photostimulation (5 s, 20 Hz) and the other of which was inactive (*Figure 6A*). Mice had ad libitum access to food during both training and testing. We then tested these mice in 150 min trials during the light phase to see whether they would engage in operant responding for AgRP neuron stimulation. Strikingly, we found that mice engaged in lever pressing in order to optically stimulate their AgRP neurons (*Figure 6B*). This lever pressing was specifically directed toward AgRP neuron self-stimulation, because it (1) was highly biased toward the active versus inactive lever (*Figure 6B,H*), (2) was greatly reduced in control mice that lacked ChR2 expression (*Figure 6H*), and (3) underwent rapid extinction when the active lever was uncoupled from the laser (*Figure 6D*). Importantly, the effectiveness of this lever pressing in stimulating AgRP neurons was confirmed by two separate measures. First, we observed reliable temporal coordination between lever pressing and food intake in mice allowed to self-stimulate: mice engaged in repeated cycles of lever pressing followed by food consumption (*Figure 6E,F*), whereas food consumption was greatly reduced when the lever was uncoupled from the laser (1.5 ± 0.2 g for active laser vs. 0.39 ± 0.07 g for inactive laser, p<0.001). Second, we observed strong induction of the activity marker Fos in AgRP neurons from mice allowed to lever press for self-stimulation, whereas no Fos was observed in otherwise identical trials in which the laser was inactivated (*Figure 6—figure supplement 1*). Thus mice will actively lever press in order to stimulate their AgRP neurons, indicating that the activity of these cells is positively reinforcing under these conditions.

We considered two hypotheses for why AgRP neuron activity might be positively reinforcing. The first is that AgRP neuron firing is intrinsically rewarding, analogous to midbrain dopamine neurons (*Corbett and Wise, 1980*). The second is that AgRP neuron activity becomes rewarding specifically in the presence of food, because it magnifies food's intrinsic positive valence. In the latter case, mice may self-stimulate their AgRP neurons for one of two reasons: in order to enhance the incentive value of the food directly in front of them during the trial (hypothesis 2a), or because of a learned positive association that developed during training when mice lever pressed for self-stimulation in the presence of food (hypothesis 2b).

We performed a series of experiments to discriminate between these hypotheses. First, we tested whether mice trained to lever press in the presence of food would self-stimulate in a trial that lacked food. We found that they did, as self-stimulation remained robust even when food was absent during the trial (*Figure 6C,H*). This indicates that the presence of food is not acutely required for the positively reinforcing effects of AgRP neuron activity.

To investigate this phenomenon further, we prepared a second cohort of laser naïve AgRP-ChR2 mice that were trained to lever press for AgRP neuron stimulation in an identical paradigm, except that food was absent during the overnight training sessions. We then tested whether these mice would lever press for self-stimulation. These mice were ad libitum fed, but food was absent during the testing. Under these conditions, we found that mice engaged in minimal lever pressing that was indistinguishable from control mice that lacked ChR2 expression (*Figure 6G,H*). This indicates that AgRP neuron stimulation is not intrinsically rewarding (hypothesis 1), but that prior experience self-

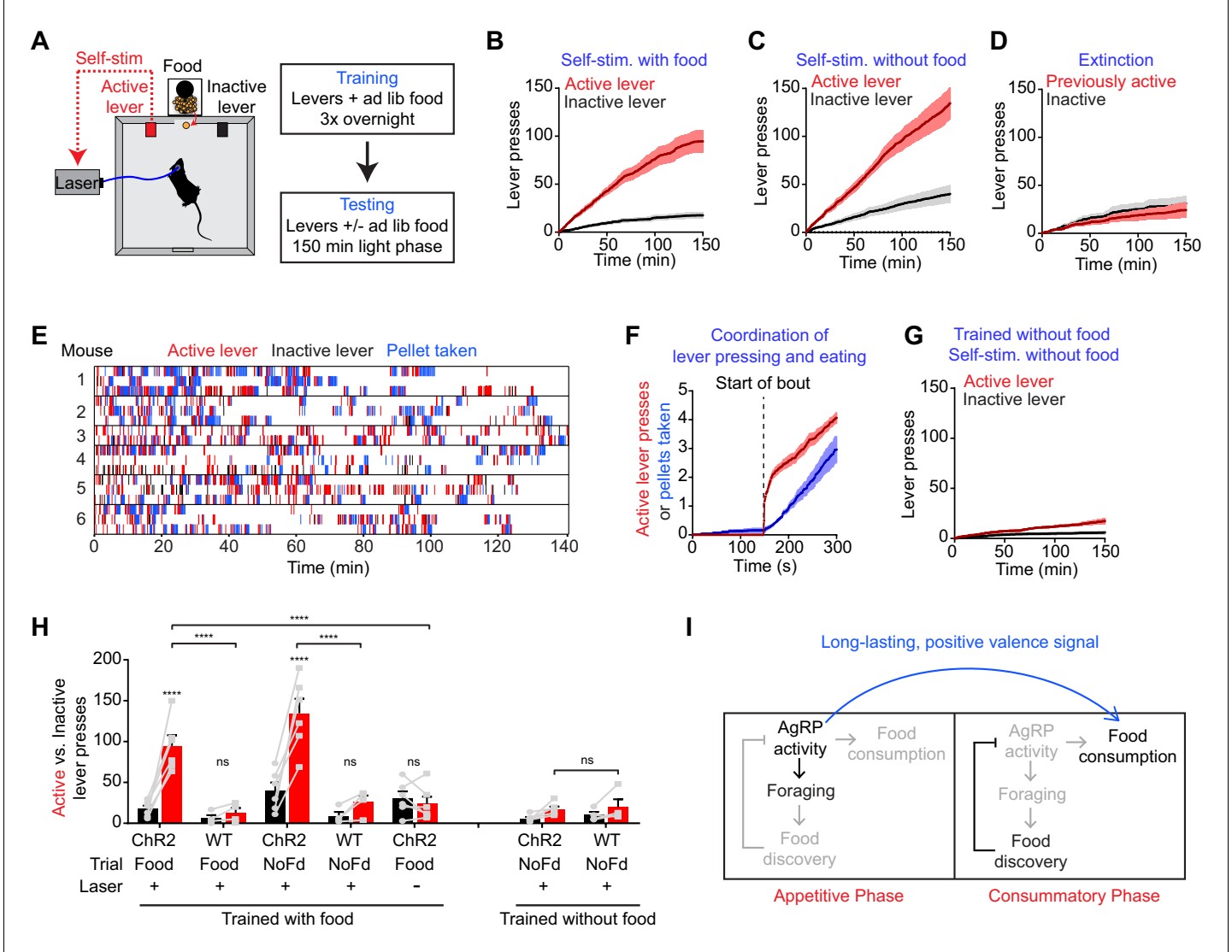

**Figure 6.** AgRP neuron activity is positively reinforcing in the presence of food. (**A**) Schematic of the positive reinforcement protocol that tests whether animals will lever press to self-stimulate AgRP neurons. (**B–D**) Plots of cumulative active (red) and inactive lever presses (black) by mice conditioned with ad lib access to food (n = 6). Filled areas indicate S.E.M. (**B**) Self-stimulation experiment with ad lib access to food pellets. (**C**) Self-stimulation experiment without access to food. (**D**) Self-stimulation experiment after extinction with ad lib access to food. (**E–F**) Temporal relationship between self-stimulation of AgRP neurons and food intake. Filled areas indicate SEM. (**E**) Raster plots of individual trials (2–3 repeats) of 6 different mice. (**F**) PSTH analysis of active lever presses and pellet consumption. Filled areas indicate SEM (n = 6). Time zero is defined as the beginning of each active lever pressing bout. A bout is defined as a lever press train segregated from other lever presses by ≥5 min. (**G**) Plots of cumulative active (red) and inactive lever presses (black) by mice conditioned without food access (n = 6). Self-stimulation experiments were conducted in the absence of food access. Filled areas indicate S.E.M. (**H**) Bar plots comparing total active (red) and inactive (black) lever presses of AgRP-ChR2 mice and WT control mice in self-stimulation experiment with ad lib (Food) or no (NoFd) access to food pellets. Asterisks on top of bar plots of active lever presses indicate significance levels compared to corresponding inactive lever presses and asterisks on top of brackets indicate significance levels for comparisons with the respective protocols, using one-way-ANOVA with Holm-Sidak's correction for multiple comparisons (****$p \leq 0.0001$, ***$0.0001 < p \leq 0.001$, **$0.001 < p \leq 0.01$, *$0.01 < p \leq 0.05$, ns $p > 0.05$). Trained with food: ChR2 food n = 6, WT food n = 4, ChR2 nofood n = 6, WT nofood n = 4; trained without food: ChR2 nofood n = 6, WT nofood n = 4. (**I**) Model for control of feeding by AgRP neurons. During the appetitive phase, AgRP neuron activity drives food seeking. The sensory detection of food silences AgRP neuron activity. However, animals still consume food during the subsequent consummatory phase because of a long-lasting, positive valence signal transmitted by AgRP neurons earlier, when food was unavailable.

The following figure supplement is available for figure 6:

**Figure supplement 1.** AgRP neurons support positive, but not negative, reinforcement.

stimulating AgRP neurons in the presence of food is sufficient for lever pressing to become positively reinforcing in food's absence (hypothesis 2b). Taken together, these data strongly support a model in which AgRP neuron activity results in a long-lasting potentiation of the rewarding properties of food. This sustained positive valence signal is sufficient to condition both Pavlovian and instrumental learning, and likely accounts for much of the behavioral response that is elicited by AgRP neuron activity.

## Discussion

AgRP neurons are a fundamental neural substrate of hunger. Nearly twenty years of investigation into the properties of these cells led to a widely accepted model for their function. The key tenets of this model were that (1) AgRP neuron activity drives feeding directly, and (2) the level of AgRP neuron activity is controlled by changes in hormones and nutrients. Both of these tenets were challenged by the recent discovery that AgRP neurons are rapidly inhibited by the sensory detection of food (*Betley et al., 2015*; *Chen et al., 2015*; *Mandelblat-Cerf et al., 2015*). Indeed, because AgRP neurons are inhibited before feeding begins, it has been unclear how these neurons are able to drive food consumption at all (*Chen and Knight, 2016*; *Seeley and Berridge, 2015*).

We hypothesized that AgRP neurons may drive feeding by transmitting a long-lasting signal that potentiates downstream circuits and persists after AgRP neuron firing has ceased (*Chen and Knight, 2016*). This would enable AgRP neuron activity before food discovery to drive feeding that occurs later, long after AgRP neurons have been silenced by sensory cues. Here we have shown that this is indeed a robust mechanism by which AgRP neurons can drive food consumption. We have shown that stimulation of AgRP neurons for as little as one minute is sufficient to increase subsequent food intake over baseline (*Figure 1F*); that this hunger signal builds up progressively over the course of 30–60 min, until prestimulated animals eat nearly as much food as mice fasted overnight (*Figure 1E*); and that this dramatic response is robust to insertion of a delay of tens of minutes between the offset of AgRP neuron stimulation and the onset of feeding (*Figure 1H*). Thus these findings explain how the remarkable behavioral effects of AgRP neurons can be reconciled with their paradoxical natural dynamics.

### Mechanisms underlying sustained hunger

It is usually assumed that a neuron driving a behavior will be most active during or immediately before the behavior's execution (*Fields et al., 2007*). For this reason, the possibility that AgRP neuron firing and feeding behavior could be separated in time by tens of minutes was unforeseen, and we are aware of few precedents in which such a vigorous and acute behavioral response can be elicited following such a long delay (*Hoopfer et al., 2015*; *Kohatsu and Yamamoto, 2015*). Consistent with this, we found no evidence for sustained behavioral effects following stimulation of an analogous population of neurons that control thirst (SFO$^{Nos1}$ neurons): for these cells, drinking behavior was tightly timelocked to the laser stimulus (*Figure 2*). Intriguingly, AgRP neuron prestimulation also strongly potentiated appetitive behaviors, since prestimulated animals were willing to perform intense lever pressing in order to obtain a food reward (*Figures 3* and *4*). Thus the sustained effects of AgRP neuron activity are not restricted to consummatory actions such as licking, chewing, and swallowing, but also extend to flexible, goal oriented behaviors associated with food obtainment. This suggests that the entire "hunger drive" that motivates food seeking and consumption is transferred to a downstream circuit node during AgRP neuron firing, such that this drive becomes independent of continued AgRP neuron activity.

The mechanisms that underlie this sustained potentiation of feeding are unknown. The fact that these effects are robust to the introduction of a 30 min delay implies that they must result from a stable change in the internal state of the mouse, rather than some feedback process that requires interaction with food. This stable change would presumably be detected as the persistent activity (or inactivity) of a population of neurons downstream of AgRP neurons within the feeding circuit. Such cells would be predicted to integrate AgRP neuron activity over time, so that they responded to rapid changes in AgRP neuron activity with a delay, thereby enabling feeding to continue after AgRP neurons have been silenced by sensory cues (*Chen and Knight, 2016*). We have shown that projections to the PVH, BNST, and LHA are each individually sufficient to drive long-lasting increases in food intake (*Figure 4*). In addition, a recent report showed that consumption of palatable foods

can be potentiated by prestimulation of AgRP neuron projections to the parabrachial nucleus (*Campos et al., 2016*). Thus the persistent orexigenic effects of AgRP neuron activity do not depend on a single circuit node.

Mechanisms for generating persistent neural activity include both cell-intrinsic processes, such as changes in membrane conductance, as well as circuit level mechanisms, such as recurrent excitatory loops (*Major and Tank, 2004*; *Wang, 2001*). These mechanisms are often invoked to explain neural processes that have a duration of seconds, such as maintenance of working memory, rather than the behavioral potentiation that persists for tens of minutes described here. Whether similar or different mechanisms underlie the control of feeding by AgRP neurons remains unknown. Addressing this question will require detailed analysis of the dynamics and physiology of relevant downstream circuit elements, which may be the direct targets of AgRP neurons (*Betley et al., 2013*; *Campos et al., 2016*; *Dietrich et al., 2012*; *Garfield et al., 2015*; *Padilla et al., 2016*) or alternatively cells that are several synapses removed from the arcuate feeding circuit (*Seeley and Berridge, 2015*).

One point not addressed by our experiments is whether optical stimulation has a long-lasting effect on the activity of AgRP neurons themselves. While we have not measured how AgRP neurons respond to our stimulation protocol in vivo, available evidence suggests that optogenetic stimulation does not result in sustained activation of these cells. This evidence includes (1) the observation from in vivo optrode recordings (*Mandelblat-Cerf et al., 2015*) that ~5 min of intermittent 20 Hz optogenetic stimulation of AgRP neurons does not result in a sustained alteration of firing (31/33 neurons returned to baseline immediately upon laser offset, and the remaining two cells within two minutes); and (2) the finding from slice recordings that 30 min of intermittent 20 Hz optogenetic stimulation does not result in a sustained increase in AgRP neuron firing in vitro (*Aponte et al., 2011*). In addition, it is important to note that the sensory detection of food can inhibit AgRP neurons even in the presence of ongoing excitatory input, such as high dose ghrelin treatment (*Chen et al., 2015*). Therefore the presentation of food would likely inhibit any residual AgRP neuron activation that persisted after photostimulation, and consequently the interpretation of the experiments described here would be largely unchanged. Nevertheless, future experiments that record AgRP neuron activity in vivo in the context of different optogenetic stimulation paradigms will further clarify this issue.

## The role of the neuropeptides NPY and AgRP in the sustained feeding response

One mechanism for the generation of persistent neural activity is the release of neuromodulators (*Major and Tank, 2004*), and AgRP neurons express two neuropeptides regulate feeding, NPY and AgRP (*Clark et al., 1985*; *Fan et al., 1997*; *Hahn et al., 1998*; *Ollmann et al., 1997*). In slice, NPY has been shown to induce long-lasting changes in membrane excitability and neurotransmitter release in certain contexts (*Dubois et al., 2012*; *Fu et al., 2004*; *Roseberry et al., 2004*). In addition, injections of NPY into the brain can drive voracious feeding with kinetics and duration that vary depending on the protocol (*Clark et al., 1985*; *Morley et al., 1987a*; *1987b*). While some studies have concluded that NPY plays a largely redundant role in the regulation of feeding (*Erickson et al., 1996*; *Krashes et al., 2013*; *Qian et al., 2002*), others have suggested it is more essential (*Bannon et al., 2000*; *Patel et al., 2006*). Whether NPY signaling participates in the sustained potentiation of feeding described here remains to be determined.

Unlike NPY, the AgRP neuropeptide can potentiate food intake for as long as two weeks in certain contexts (*Hagan et al., 2000*; *Krashes et al., 2013*; *Nakajima et al., 2016*). However, we believe that AgRP is unlikely to mediate the behavioral responses described here, for two reasons. First, the behavioral responses we observe following AgRP neuron prestimulation are almost immediate, in that animals begin to eat within seconds of food presentation (*Figure 1*). By contrast the release of the AgRP neuropeptide requires at least two hours to affect feeding (*Krashes et al., 2013*). Thus the AgRP neuropeptide appears to act too slowly to explain our findings. Second, our projection stimulation experiments show that AgRP neuron projections to the PVH and BNST are both efficient in driving sustained feeding (*Figure 4*). However AgRP neuron projections to the BNST have been shown to function by targeting BNST neurons that do not express the melanocortin 4 receptor (MC4R), which is the target of AgRP (*Garfield et al., 2015*). This implies that the AgRP neuropeptide cannot be the molecule that drives feeding in our ARC → BNST stimulation experiments, and therefore that other neurotransmitters released by these cells (GABA or NPY) must underlie their sustained effects.

# AgRP neurons drive feeding through a sustained positive valence mechanism

Food seeking and consumption are motivated behaviors. An important and unresolved question regards the nature of the motivational processes that AgRP neurons engage in order to promote feeding. Traditionally, motivational valence has been assigned by measuring the behavioral response to ongoing neural stimulation (*Fields et al., 2007*; *Kravitz et al., 2012*; *Namburi et al., 2015*). However, the discovery that AgRP neurons are rapidly inhibited by the sensory detection of food (*Chen et al., 2015*) and consequently drive food intake through a long-lasting, persistent mechanism (*Figures 1*, *3,* and *4*) implies that the motivational signals most relevant for food consumption are those that persist after AgRP neuron firing has ceased. These motivational signals have never been investigated.

To explore the properties of these long-lasting motivational cues, we stimulated AgRP neurons before food availability and then measured how this prestimulation affected the preference for subsequently presented foods. We found that prior AgRP neuron stimulation robustly conditioned flavor and food preference (*Figure 5*). This effect was sufficient to motivate mice following a single trial to overeat a test food that had been paired with AgRP neuron prestimulation (*Figure 5B,C*). Importantly, this conditioned appetite was specific to the food paired with AgRP neuron prestimulation, because it could be blocked by ad libitum access to the paired food but not a different, unpaired food (*Figure 5B*). A similar phenomenon, known as 'conditioned craving', has been observed in rats that are fed a specific kind of pellet only when food deprived (*Petrovich et al., 2007*). The finding that this food-specific craving can be trained by AgRP neuron prestimulation argues that these neurons motivate feeding by potentiating the incentive salience or perceived rewarding properties of food encountered during a state of energy deficit (*Seeley and Berridge, 2015*). Of note, the idea that food deprivation can magnify food reward has long been recognized as a critical mechanism that drives feeding (*Berridge, 2004*; *Cabanac, 1971*), but the underlying neural mechanisms have been unclear. We propose that AgRP neurons are the origin of this effect.

A prediction of this positive valence model is that animals should engage in operant responding in order to stimulate AgRP neuron activity. We found that this is indeed the case, as animals will actively lever press in order to turn on (*Figure 6*) but not in order to turn off (*Figure 6—figure supplement 1*) AgRP neuron firing. Importantly, this instrumental responding required either that food was present during the trial (*Figure 6B*) or that animals had previously been allowed to self-stimulate in the presence of food, in order to learn this positive association (*Figure 6C*). This argues that AgRP neuron activity is not necessarily intrinsically rewarding, but that it attains positive valence specifically in the presence of food. This observation is again most readily explained by a model in which AgRP neurons enhance food's intrinsically rewarding properties. By contrast, these findings do not support a model in which AgRP neurons motivate food consumption primarily through a negative valence signal (*Betley et al., 2015*), since animals do not self-administer aversive stimuli.

How can we reconcile our data with a prior report that AgRP neuron activity has negative valence? That study measured the valence of ongoing AgRP neuron firing in the absence of food and concluded that it was aversive, since animals avoided places and flavors associated with elevated AgRP neuron activity (*Betley et al., 2015*). In contrast, we have measured here the valence that persists after AgRP neuron activity has ceased, because this corresponds to the natural activity pattern of these cells during food consumption (*Chen et al., 2015*). This has revealed that AgRP neurons transmit a previously unsuspected positive valence signal that can robustly condition appetite (*Figure 5*) and motivate instrumental responding (*Figure 6*). Therefore, these data support an important role for a long-lasting, positive valence mechanism by which AgRP neurons motivate food consumption. Nevertheless, these arguments do not rule out an additional role for a negative valence signal that functions primarily prior to food discovery and contributes to food seeking or learning (*Betley et al., 2015*). Indeed, there is evidence that both positive and negative valence mechanisms contribute to the control of feeding behavior (*Berridge, 2004*; *Bindra, 1976*; *Fulton, 2010*; *Hull, 1943*; *Lockie and Andrews, 2013*). Investigation of the neural circuitry downstream of AgRP neurons may provide additional insight into how these parallel mechanisms are coordinated.

## Optogenetic replay of the natural dynamics of AgRP neurons

Optogenetics enables selective manipulation of genetically defined cell types and thereby determination of their causal role in behavior (*Adamantidis et al., 2015*). An assumption implicit in most optogenetic experiments is that the pattern of artificial stimulation approximates the natural firing pattern of the cells, at least in its key features: otherwise, the relevance of any optically-elicited behavior is unclear. While this caveat is widely understood, the lack of information about the in vivo dynamics of many cell types has often precluded consideration of their natural firing patterns. For AgRP neurons, it was long assumed that these cells are highly active during feeding and only inhibited following food consumption, and this model guided the design of early optogenetic studies. However the discovery that AgRP neurons are inhibited by the sensory detection of food, and therefore have a firing pattern during feeding that is the opposite of what was believed, calls for reinvestigation of how these cells control behavior. Here we have explored this question by using a prestimulation protocol that mimics the broad features of the natural dynamics of AgRP neurons. Using this new stimulation protocol, we are able to reconcile the paradoxical dynamics of AgRP neurons with their well-established function to promote feeding; identify novel mechanisms by which these neurons motivate behavior; and raise new questions about the downstream feeding circuit that await investigation.

## Materials and methods

### Mice

Mice were group housed on a 12:12 light:dark cycle with ad libitum access to water and mouse chow (PicoLab Rodent Diet 20, 5053 tablet, TestDiet). Adult mice (8–16 weeks old) were used for all experiments. For channelrhodopsin-2 expression in AGRP neurons, *Agrp-IRES-Cre* mice (Jackson Labs Stock 012899, *Agrp*$^{tm1(cre)Lowl}$/J) were crossed with Ai32: *ROSA26-loxStoplox-ChR2-eYFP* (Jackson Labs stock 012569, B6;129S-*Gt(ROSA)26Sor*$^{tm32(CAG-COP4*H134R/EYFP)Hze/J}$) to generate double mutant animals. Wildtype C57BL/6J mice were used as controls. No statistical methods were used to determine sample sizes. Experimental protocols were approved by the University of California, San Francisco IACUC (Protocol AN133011) following the NIH guidelines for the Care and Use of Laboratory Animals.

### Stereotaxic viral delivery and fiber implant

Recombinant AAV expressing ChETA$_{TC}$ (AAV5-EF1α-DIO-hChR2(E123T/T159C)-2A-mCherry-WPRE) was purchased from the UNC Vector Core. AAV was stereotaxically injected into the SFO of NOS1-IRES-Cre (Jackson Labs Stock 017526, *Nos1*$^{tm1(Cre)Mgmj}$/J) mice at 0.55 mm (A/P), −2.75 mm (D/V), 0 mm (M/L) relative to bregma.

Custom-made fiberoptic implants (0.39 NA Ø200 µm core Thorlabs FT200UMT and CFLC230-10) were installed above the SFO (bregma: AP: 0.55 mm, DV: 2.45 mm, ML: 0 mm), the ARC (bregma: AP: −1.75 mm, DV: dorsal surface −5.6 mm, ML: −0.25 mm), PVH (bregma: AP:−0.75 mm, DV: -4.3 mm, ML: −0.2 mm), BNST (bregma: AP: +0.5 mm, DV: −4.2 mm, ML: −0.55 mm) or LHA (bregma: AP: −1.4 mm; ML: −1.2 mm; DV: −4.7 mm).

### Optogenetic stimulation

A 473 nm laser was modulated by Coulbourn Graphic State software through a TTL signal generator (Coulbourn H03-14) and synchronized with behavior experiments. The laser was split through a 4-way splitter (Fibersense and Signals) or passed through a single patch cable (Doric Lenses). The laser was then passed to custom-made fiber optic patch cables (Thorlabs FT200UMT, CFLC230-10; Fiber Instrument Sales F12774) through a rotary joint (Doric Lens FRJ 1x1). Patch cables were connected to the implants on mice through a zirconia mating sleeve (Thorlabs ADAL1). For opto-stimulation protocols, laser was modulated at 20 Hz on a 2 s ON and 3 s OFF cycle with 1 ms pulse width unless otherwise specified. Laser power was set within 15–20 mW at the terminal of patch cable unless otherwise specified. We estimated the light power at the ARC, PVH and BNST to be 4.02, 4.02 and 9.4 mW/mm$^2$ respectively. Effective power is likely lower due to loss at the cable-implant connection.

## Functional evaluation of fiber placement

At the end of experiments, each AgRP-ChR2 mouse was further tested with a positive control protocol (60 min laser stimulation during food availability) to confirm correct fiber optic placement. Two AgRP-ChR2$^{PVH}$ and two AgRP-ChR2$^{LHA}$ mice that displayed a less than 20% increase of food intake during this positive control protocol were excluded.

## Pre-stimulation evoked food intake

Mice were allowed to recover for seven days after implant surgery before experiments. In addition to regular chow, mice were supplied *ad libitum* with the food pellets used during testing (20 mg Bio-Serv F0163) in their home cage unless otherwise specified. Mice were habituated to the behavioral chambers (Coulbourn H10-11M-TC with H10-11M-TC-NSF) and pellet dispensing systems (Coulbourn H14-01M-SP04 and H14-23M) for three days before the first experiment. Mice were provided ad libitum access to food and water unless otherwise specified and tested during the early phase of the light cycle. All pre-stimulation food intake experiments follow this general structure: 70 min habituation/pre-stim period with no food access followed by 60 min food access. Pellet removal was detected using a built-in photo-sensor (Coulbourn H20-93). Food pellets left on the ground after each session were counted and deducted from the total food consumed.

To test whether stimulation of AgRP neuron soma or axonal terminals in the PVH, BNST, and LHA induces food intake, each mouse was tested in the following sequence of experiments on consecutive days: 1- 1- 2- 1- 2- 1- 1- 3- 1- 3- 1 (protocols are described in the table below). All mice were naïve (never stimulated by a laser previously) on the first day of these tests.

| Protocol 1 | 70 min habituation | | 60 min food access |
|---|---|---|---|
| Protocol 2 | 10 min habituation | 60 min opto-stim | 60 min food access |
| Protocol 3 | 60 min opto-stim | 10 min habituation | 60 min food access |
| Positive control | 70 min habituation | | 60 min food access with opto-stim |

To examine the relationship between stimulation protocols and induced food intake, mice were tested with the following protocols once each in semi-randomized order:

| 69 min habituation | 1 min opto-stim | | 60 min food access |
|---|---|---|---|
| 65 min habituation | 5 min opto-stim | | 60 min food access |
| 55 min habituation | 15 min opto-stim | | 60 min food access |
| 40 min habituation | 30 min opto-stim | | 60 min food access |
| 10 min habituation | 60 min opto-stim | | 60 min food access |
| 10 min habituation | 30 min opto-stim | 30 min habituation | 60 min food access |
| 30 min habituation | 30 min opto-stim | 10 min habituation | 60 min food access |
| 30 min habituation | 30 min opto-stim (10 Hz; 4s ON, 1s OFF) | | 60 min food access |
| 30 min habituation | 30 min opto-stim (20 Hz; 2s ON, 8s OFF) | | 60 min food access |

## Lickometer assay

Mice were habituated to the optical lickometer (Coulbourn H24-01M, H20-93) at least a week prior to experiments. Behavioral experiments were performed during the light cycle using the protocols described below.

| pre-stimulation | 70 min habituation no water access | | 30 min water access |
|---|---|---|---|
| no stimulation | 10 min habituation no water access | 60 min stimulation no water access | 30 min water access |

| pre-stimulation + co-stimulation | 45 min habituation + water access | 30 min stimulation no water access | 30 min stimulation + water access |
| --- | --- | --- | --- |
| co-stimulation | 45 min habituation + water access | | 30 min stimulation + water access |

During behavioral testing of SFO$^{NOS1}$::ChETA$_{TC}$ mice, water access was prevented using a custom-made lickometer blocker. Co-stimulation data were based on experiments performed in (*Zimmerman et al., 2016*).

### Progressive ratio testing

For training, mice were acutely food deprived 5 hr before the start of dark cycle and trained with fr1 and fr7 protocols overnight until active lever presses exceeded 200. Mice were then acutely food deprived 5 hr before the start of the dark cycle and trained with progressive ratio 3 (PR3) task for 1.5 hr.

During the first 70 min of the testing protocol (habituation/pre-stim), access to the levers and pellet trough was blocked using a custom-cut acrylic board. At 70th minute of the protocol, the acrylic board was removed and a single pellet was delivered following pressing the active lever according to a PR3 schedule. Each experiment was repeats 2–7 times; no-stim and pre-stim are repeated the same number of times for each mouse.

### Conditioned appetite assay

All mice were naïve (never stimulated by a laser previously) on the first day of these tests. Mice were provided with ad libitum regular chow and without any test pellets in their homecage from the beginning of the test unless otherwise specified. The regular chow was PicoLab Rodent Diet 20 (5053), which has an energy density of 3.43 kcal/g and macronutrient composition of approximately 21.0%:5.0%:53.4% Protein:Fat:Carbohydrate. The test pellets were BioServ Dustless Precision Pellets, which have an energy density of 3.35 kcal/gram and macronutrient composition of approximately 21.3%:3.8%:54%. The regular chow was formulated as an oval pellet of approximately 3/8 × 5/8 × 1 inch, whereas the test pellets were formulated as a much smaller, smooth round pellet (20 mg).

Each mouse was tested in the following sequence of experiments on consecutive days: 1- 1- 2- 1- 2- 1- 2–1 (protocols are described above in the pre-stimulation session). Protocol 2 with pre-stimulation was considered as a conditioning trial in this experiment. At the end of the 7th experiment or the third conditioning trial, *ad lib* amount of test pellets were put into the mice home cage in order to induce extinction of previously conditioned appetite.

### Conditioned flavor preference assay

All mice were naïve (never stimulated by a laser previously) on the first day of these tests. Mice were first habituated to two differently flavored non-nutritive gels (Hunt's Snack Pack Sugar Free Strawberry & Orange) that were sweetened with sucralose. Mice were transferred to a clean cage after initial habituation without test gel. Baseline flavor preference was determined in two separate food choice assays conducted in two consecutive days. Each food choice assay consists of 30 min habituation and 15 min food consumption. In the next four days, the following two protocols were used to condition the mice to their less preferred flavor with orders inverted each day.

| 30 min pre-stim | 30 min consumption of less preferred gel (0.3 g provided) |
| --- | --- |
| 30 min nostim | 30 min consumption of preferred gel (0.3 g provided) |

On the day following the last conditioning session, two food choice assays separated by 4 hr were conducted to determine the conditioned tasted preference.

## Self-stimulation

Mice were initially habituated to the operant chamber and test pellets in the same way as mice in pre-stimulation evoked food intake assay. Each cohort was semi-randomly split into two groups (group A and group B). Of note, all mice were naïve to the lever (never exposed to a lever before) and to the laser stimulation on the first day of these tests. The lever on one side of the operant chamber was semi-randomly assigned to each mouse as the active lever. The lever location was counterbalanced within each group. The spatial localization of active lever and inactive lever was fixed for each mouse through the whole experiment.

Group A mice were initially tested with the following protocols to determine baseline lever pressing during the light phase:

| Food availability | Experiment duration | Lever-laser pairing | Repeats |
| --- | --- | --- | --- |
| *ad lib* food pellet access | 2.5 hr | off | 1 |

Group A mice were then habituated overnight with the following protocol:

| Food availability | Experiment duration | Lever-laser pairing | Repeats |
| --- | --- | --- | --- |
| *ad lib* food pellet access | overnight | on | 3 |

After conditioning, group A mice were then tested with the following protocol during the light phase:

| Food availability | Experiment duration | Lever-laser pairing | Repeats |
| --- | --- | --- | --- |
| *ad lib* food pellet access | 2.5 hr | on | 2–3 |
| no food access | 2.5 hr | on | 2–3 |

To test memory extinction, group A mice were then conditioned with the following protocol:

| Food availability | Experiment duration | Lever-laser pairing | Repeats |
| --- | --- | --- | --- |
| *ad lib* food pellet access | 2.5 hr | off | 3–4 |

The data from the last trial of the extinction experiments were compared to the pre-extinction trials in the analysis.

Group B mice were initially tested with the following protocols to determine baseline lever press during the light phase:

| Food availability | Experiment duration | Lever-laser pairing | Repeats |
| --- | --- | --- | --- |
| no food pellet access | 2.5 hr | off | 1 |

Group B mice were then habituated overnight with the following protocol:

| Food availability | Experiment duration | Lever-laser pairing | Repeats |
| --- | --- | --- | --- |
| no food pellet access | Overnight | on | 3 |

After conditioning, group B mice were then tested with the following protocol during the light phase:

| Food availability | Experiment duration | Lever-laser pairing | Repeats |
|---|---|---|---|
| no food pellet access | 2.5 hr | on | 2–3 |

## Negative reinforcement assay

AgRP-ChR2 mice with optical implants above the ARC were used in this experiment. Mice were naive to the lever (never exposed to a lever before) at the beginning of this experiment. The spatial localization of the active lever and inactive lever in the cage was counterbalanced within each cohort and fixed through the whole experiment.

Mice were then conditioned to active lever during the beginning of the dark phase for 2 hr for 3–4 times. During conditioning, each mouse received constant 20 Hz laser stimulation that could be turned off for 20 s by each press of the active lever. After three repeats of this conditioning protocol, mice were then tested with the same protocol during the light phase. Each trial lasted for 1 hr.

## Fos staining following self-stimulation

Mice were tested with the self-stimulation protocol described above in the presence of food for 2.5. Immediately after the self-stimulation experiment, each mouse was perfused transcardially with PBS buffer followed by formalin. Brains were removed, postfixed in 4% PFA and transferred to PBS buffered 20% sucrose. Free floating sections (40 µm) were prepared with a cryostat, blocked (3% BSA, 2% NGS, and 0.1% Triton-X in PBS for 2 hr), and then incubated with primary antibody (chicken anti-GFP, Abcam, ab13970, 1:1000; goat anti-Fos, Santa Cruz, SC52G, 1:500) overnight at 4°C. Samples were washed, incubated with secondary antibody (goat anti-chicken Alexa 488 secondary antibody; Invitrogen, 1:1000; donkey anti-goat Alexa 568 secondary antibody; Invitrogen, 1:1000) for 2 hr at room temperature, washed, mounted, and imaged with a confocal microscope. Images for direct comparison are imaged with the same settings.

To quantify the percentage of AgRP neurons that express Fos, we first identified 100 putative AgRP cells from each mouse based on GFP fluorescence using the ImageJ Cell Counter Plugin. We then manually quantified the presence or absence of Fos staining in those previously defined cells.

## Immunofluorescence

Immunofluorescence was performed as previously described (*Chen et al., 2015*) using the following antibodies: Chicken anti-GFP (Aves Lab, GFP-1020, 1:1000); Goat anti-chicken Alexa-fluorophore 488 (Life Technologies A11039, 1:1,000).

## Statistics

Raw behavioral data were analyzed with custom MATLAB scripts. Multiple measurements from the same mouse in the same experiment (e.g. on different days) were considered technical repeats and were averaged before statistical analysis. The average of these technical repeats for each mouse in each experiment was considered a single biological repeat and was used to determine sample size for statistical analysis. Data were analyzed by two-way ANOVA using Graphpad Prism 6 to test for an effect of genotype and stimulation protocol (experiments with WT control) or one-way ANOVA (experiments without WT control). Individual p-values were corrected using Holm-Sidak's multiple comparison test. Regression analysis for experiments investigating feeding kinetics was performed using Graphpad.

## Acknowledgements

We thank Vanessa Ruta, Nirao Shah, Jennifer Garrison, Evan Feinberg, Anatol Kreitzer, and Vikaas Sohal for helpful discussions. ZAK is a New York Stem Cell Foundation-Robertson Investigator. YC is an HHMI International Student Research Fellow. CAZ is an NSF Predoctoral Fellow and UCSF Discovery Fellow.

## Additional information

### Funding

| Funder | Grant reference number | Author |
|---|---|---|
| National Institute of Diabetes and Digestive and Kidney Diseases | R01DK106399 | Zachary A Knight |
| National Institute of Neurological Disorders and Stroke | R01NS094781 | Zachary A Knight |
| New York Stem Cell Foundation | Robertson Investigator Award | Zachary A Knight |
| American Diabetes Association | Pathway Accelerator Award | Zachary A Knight |
| Rita Allen Foundation | Rita Allen Scholar | Zachary A Knight |
| Alfred P. Sloan Foundation | | Zachary A Knight |
| McKnight Endowment Fund for Neuroscience | | Zachary A Knight |
| National Institute of Diabetes and Digestive and Kidney Diseases | P30DK098722 | Zachary A Knight |
| National Institute of Diabetes and Digestive and Kidney Diseases | P30DK063720 | Zachary A Knight |
| Brain and Behavior Research Foundation | | Zachary A Knight |
| National Institute of Diabetes and Digestive and Kidney Diseases | DP2DK019342 | Zachary A Knight |
| UCSF Program for Breakthrough Biological Research | | Zachary A Knight |

The funders had no role in study design, data collection and interpretation, or the decision to submit the work for publication.

### Author contributions

YC, Y-CL, CAZ, RAE, Conception and design, Acquisition of data, Analysis and interpretation of data, Drafting or revising the article; ZAK, Conception and design, Analysis and interpretation of data, Drafting or revising the article

### Author ORCIDs

Zachary A Knight, http://orcid.org/0000-0001-7621-1478

### Ethics

Animal experimentation: This study was performed in strict accordance with the recommendations in the Guide for the Care and Use of Laboratory Animals of the National Institutes of Health. All experiments were approved by the UCSF institutional animal care and use committee (IACUC protocol AN133011).

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
