## [Decision Letter]

Thank you for submitting your article "Hunger neurons drive feeding through a sustained, positive reinforcement signal" for consideration by *eLife*. Your article has been reviewed by two peer reviewers, including Michael Krashes (Reviewer #3), and the evaluation has been overseen by a Reviewing Editor and a Senior Editor.

The reviewers have discussed the reviews with one another and the Reviewing Editor has drafted this decision to help you prepare a revised submission.

Summary:

This manuscript by Chen et al. posits that AgRP neurons promote feeding through a sustained, positive reinforcement signal, as opposed to a previously described mechanism in which AgRP neurons signal via negative reinforcement. Overall these studies are expertly designed and carried out and the conclusions deduced from the results do indeed support a model of positive reinforcement. Additionally the data is presented in a way that makes it very easy for the reader to follow.

Essential revisions:

1) The phenomenology is interesting, but a main claim is that this optogenetic stimulation is more physiological and many aspects of the physiology of these neurons is not recapitulated by this stimulus – for example the fact that these neurons reactivate if the animal detects food that turns out to have no calories and the fact that optogenetic stimulation is synchronous are major experimental confounds.

Although the vast majority of AgRP optogenetic studies employ 20 Hz stimulation protocols to elicit feeding behaviors, it should be discussed that this frequency is likely supra-physiological as Mandelblat-Cerf et al., 2015 report that mean spike frequency of AgRP neurons during the AM is about 1.4 Hz and PM is about 7.6 Hz. This should be discussed as possibility for the observations. It is realized that the initial finding of prestim evoking food intake occurs even at 10 Hz photostimulation (Figure 1—figure supplement 1) but the remainder of the studies are done at the higher frequency.

2) The authors hypothesize that plasticity changes are occurring in downstream targets of AgRP neurons, but it is possible that 1 hr of synchronized, high frequency AgRP stimulation may induce some form of plasticity in AgRP neurons themselves whereby even after photostimulation ends, the firing of these neurons continues. It would be insightful to either eliminate this possibility or demonstrate this through combinatorial photometry/optogenetics. However, given the difficulty of such an experiment, a mention of this possibility would be sufficient.

The fact that NPY causes delayed feeding (and so does AgRP) is well known, therefore, it is not surprising that activation of AgRP neurons leads to sustained feeding even after a delayed interval to present food. In addition, it is not clear that the approach used (optogenetic stimulation) is altering the excitability of the neurons beyond the period of light exposure. It would be important to record AgRP neuron activity here to understand whether these neurons are switching off or not after ceasing light exposure. Other approaches and controls should be used, including AgRP neuronal inhibition.

3) For the conditioned appetite assay (Figure 5), more information should be provided about the differences in the test pellets versus home cage chow. The authors note the difference in size, shape and texture between the two food sources but don't elaborate on those specific distinctions or any differences in the nutritive content. This should be explained clearly since similar/identical caloric content (same proportion of carbs/fat/protein) of the two food sources make this result even stronger. Additionally, the authors refer to the "taste or texture" of the test pellets and hypothesize that it may be these specific sensory properties that reflects the attribution of incentive value, making a fuller description of these differences of the two food sources imperative.

4) The results described in Figure 1–Figure 4 are not surprising and they support many other recent work from different groups (Lowell, Horvath, Palmiter, and Sternson) showing that AgRP neurons control appetitive behaviors via a complex network of neuronal projections. The mechanisms underlying the delayed effects of AgRP neuron activation on feeding should be more fully discussed in the paper.

5) The authors refer to 3 articles that evaluated the activity of AgRP neurons in vivo. Upon review, two of these articles used calcium imaging, which is an indirect way to infer neuronal activity. The only manuscript that directly accessed AgRP neuronal activity using in vivo electrophysiology shows that ~1/3 of AgRP neurons do not show a decrease in activity upon presentation of food/feeding. Here, the authors hit the whole population of AgRP neurons. Based on these facts, it is not clear why the authors call this a 'more physiologic stimulation protocol'.

6) Figure 5 and Figure 6 are the most interesting and novel, bringing relevant information to the field in apparent contradiction with the experiments described in Betley et al. 2015. However, experiments to characterize the activity of these neurons in the Betley et al. 2015 paper are ignored in this manuscript and not cited. For example, in the Discussion (subsection “AgRP neurons drive feeding through positive reinforcement”), which states that Betley et al. "measured the valence of ongoing AgRP neuron activity," they also measured the valence of silencing the natural activity patterns of AgRP neurons in hunger and found the same results from bidirectional experiments.

7) Finally, the direct comparisons to Betley et al. paper should be tempered because although the conclusions are different, so are the experiments employed to make their conclusions.

---

## [Author Response]

*Essential revisions:*

*1) The phenomenology is interesting, but a main claim is that this optogenetic stimulation is more physiological and many aspects of the physiology of these neurons is not recapitulated by this stimulus – for example the fact that these neurons reactivate if the animal detects food that turns out to have no calories and the fact that optogenetic stimulation is synchronous are major experimental confounds.*

We agree that our protocol does not recapitulate all of the aspects of the natural firing of AgRP neurons, such as the fact that their activity is asynchronous. This is also true of previous optogenetic studies. Nevertheless, our protocol is “more physiological” because it mimics a critical element of the natural firing pattern of these neurons that has been absent from prior studies: their rapid inhibition upon the sensory detection of food. We are careful to state in several places that our protocol mimics only the “broad features” of the natural activity of these cells, not all aspects.

*Although the vast majority of AgRP optogenetic studies employ 20 Hz stimulation protocols to elicit feeding behaviors, it should be discussed that this frequency is likely supra-physiological as Mandelblat-Cerf et al., 2015 report that mean spike frequency of AgRP neurons during the AM is about 1.4 Hz and PM is about 7.6 Hz. This should be discussed as possibility for the observations. It is realized that the initial finding of prestim evoking food intake occurs even at 10 Hz photostimulation (Figure 1—figure supplement 1) but the remainder of the studies are done at the higher frequency.*

Our optogenetic stimulation protocol uses a 2 second ON, 3 seconds OFF structure. Thus our “20 Hz” stimulation protocol stimulates the neurons at an average rate of 20 Hz multiplied by 2/5 = 8 Hz. This is very close to the 7.6 Hz firing rate that Mandelblat-Cerf et al. report for AgRP neurons in the PM. It is also lower than the ~20 Hz firing rate that Mandelblat-Cerf et al. report for AgRP neurons from fasted mice. Thus our stimulation frequency is not supraphysiological. We also show that the 2:3 ON:OFF pattern of optogenetic stimulation is not critical for our results, and that other stimulation patterns utilizing more tonic activation yield very similar behavioral responses (Figure 1—figure supplement 1). We have adjusted the text to make this point clearer.

*2) The authors hypothesize that plasticity changes are occurring in downstream targets of AgRP neurons, but it is possible that 1 hr of synchronized, high frequency AgRP stimulation may induce some form of plasticity in AgRP neurons themselves whereby even after photostimulation ends, the firing of these neurons continues. It would be insightful to either eliminate this possibility or demonstrate this through combinatorial photometry/optogenetics. However, given the difficulty of such an experiment, a mention of this possibility would be sufficient.*

This is a good point, and we discuss this possibility at length in the revised text (Discussion section, subsection “Mechanisms underlying sustained hunger”). We think it is unlikely that continued elevated firing of AgRP neurons after the cessation of photostimulation explains our behavioral responses, for three reasons.

1. We observed significant elevation of food intake following as little as 1 or 5 minutes of prestimulation. Mandelblat-Cerf et al. performed optrode recordings from photostimulated AgRP neurons in vivo, and showed that the activity of almost all of these neurons (31/33 cells) returns to baseline after photostimulation of approximately 5 minutes. Therefore plasticity at AgRP neurons themselves does not appear to develop in vivo following photostimulation of this duration.

2. It has been reported that photostimulation of AgRP neurons in slice for 30 minutes does not result in an elevated firing rate after the cessation of stimulation. Instead there is a slight decrease in action potential probability. See Supplementary Figure 1 from (Aponte et al., 2011)

3. The sensory detection of food strongly inhibits AgRP neurons. For example, food presentation rapidly reverses the activation of AgRP neurons induced by pharmacologic doses of ghrelin (Chen et al., 2015). Therefore, the presentation of food should silence any AgRP neuron activity that remains elevated following photostimulation. In this scenario, the interpretation of most experiments in this paper would be unchanged.

*The fact that NPY causes delayed feeding (and so does AgRP) is well known, therefore, it is not surprising that activation of AgRP neurons leads to sustained feeding even after a delayed interval to present food.*

In the revised manuscript we discuss the results of studies in slice showing that NPY can have durable effects on neural excitability or neurotransmitter release. We also cite studies that injected NPY into the brain and measured effects on feeding, as well as studies from NPY knockout mice. We agree that all of these results are interesting and important. However they do not anticipate any of the findings we report here, nor do they reveal the mechanism by which AgRP neurons drive sustained feeding.

We also discuss studies reporting delayed effects of the AgRP neuropeptide on feeding and explain why these AgRP-peptide dependent effects are unlikely to account for the behavioral response we observe. The key difference is that the delayed feeding caused by the AgRP peptide develops over hours and is smaller in magnitude, but persists for days, whereas the phenomenon we describe is immediate, intense, and persists for tens of minutes.

We disagree that it is “not surprising” that prestimulation of AgRP neurons causes the acute and dramatic feeding response we observe. Several studies have previously described the optogenetic stimulation of AgRP neurons, and these studies invariably reported that when the laser shuts off, the mouse stops eating (Aponte et al., 2011; Atasoy et al., 2012; Steculorum et al., 2016). This observation has been logically interpreted to imply that continued stimulation of AgRP neurons is necessary for the acute feeding response. This conclusion is consistent with the general assumption that a neuron driving a behavior should fire during or immediately preceding the behavior’s execution. However we show that this is not the case for AgRP neurons. Instead, the behavioral response can be elicited by prestimulation that occurs tens of minutes before the food is presented. This has never before been shown and is quite surprising. Indeed, we show that an analogous population of thirst neurons do not have this property (Figure 2).

*In addition, it is not clear that the approach used (optogenetic stimulation) is altering the excitability of the neurons beyond the period of light exposure. It would be important to record AgRP neuron activity here to understand whether these neurons are switching off or not after ceasing light exposure. Other approaches and controls should be used, including AgRP neuronal inhibition.*

Please see the first response to comment 2) above regarding the excitability of AgRP neurons. We have included an extensive discussion of this point in the revised manuscript.

We agree that neuronal inhibition experiments could provide additional mechanistic information about the sustained effects of AgRP neuron activity on feeding. However we have been unable to inhibit food intake by optogenetic silencing of AgRP neurons, despite extensive efforts in our lab to develop such protocols (e.g. using Arch). We have been told by others in the field that the inability of optogenetic silencing of these neurons to inhibit feeding is a common observation and the reasons are unknown. Testing our model requires the temporal resolution of optogenetics, because we explicitly distinguish between the function of AgRP neuron activity before and after food presentation.

*3) For the conditioned appetite assay (Figure 5), more information should be provided about the differences in the test pellets versus home cage chow. The authors note the difference in size, shape and texture between the two food sources but don't elaborate on those specific distinctions or any differences in the nutritive content. This should be explained clearly since similar/identical caloric content (same proportion of carbs/fat/protein) of the two food sources make this result even stronger. Additionally, the authors refer to the "taste or texture" of the test pellets and hypothesize that it may be these specific sensory properties that reflects the attribution of incentive value, making a fuller description of these differences of the two food sources imperative.*

We have added an extensive description of the two food sources in the methods, and noted in the main text that they have very similar energy density. We thank the reviewer for pointing this out.

*4) The results described in Figure 1–Figure 4 are not surprising and they support many other recent work from different groups (Lowell, Horvath, Palmiter, and Sternson) showing that AgRP neurons control appetitive behaviors via a complex network of neuronal projections. The mechanisms underlying the delayed effects of AgRP neuron activation on feeding should be more fully discussed in the paper.*

See the second response to point 2) above. In the revised manuscript we have expanded our discussion of the possible mechanisms underlying the sustained effects of AgRP neuron activation on feeding. This section of the Discussion now totals 1162 words.

*5) The authors refer to 3 articles that evaluated the activity of AgRP neurons* in vivo*. Upon review, two of these articles used calcium imaging, which is an indirect way to infer neuronal activity. The only manuscript that directly accessed AgRP neuronal activity using* in vivo *electrophysiology shows that ~1/3 of AgRP neurons do not show a decrease in activity upon presentation of food/feeding. Here, the authors hit the whole population of AgRP neurons. Based on these facts, it is not clear why the authors call this a 'more physiologic stimulation protocol'.*

See first response to 1) above. The optogenetic protocol we describe here is more physiologic because it replicates a critical feature of the natural dynamics of these cells that has been absent from previous studies, namely the rapid silencing of most AgRP neurons upon the sensory detection of food. Previous optogenetic studies used a stimulation protocol that was the opposite of the natural firing pattern (neurons OFF before food presentation and then neurons ON after food presentation).

It is true that many caveats apply to optogenetic experiments. Our claim is only that the optogenetic protocol here mimics the natural activity pattern more closely than previous optogenetic protocols. We are careful to state in the text that our protocol recapitulates only the “broad features” of the natural dynamics of AgRP neurons, not that it mimics every aspect, which is obviously impossible.

In addition to this general point, we do not agree with the premise that the Mandelbrot-Cerf et al. electrophysiologic recordings are necessarily a more accurate measurement of AgRP neuron activity than the Betley et al. results based on calcium imaging. We discuss the reasons for this here (Chen and Knight, 2016), but the important point is that Mandelblat-Cerf et al. used a head-fixed preparation with a liquid diet, which may reduce the anticipatory response compared to freely behaving animals provided with solid food. This debate will likely continue for some time. Nevertheless, the key point is that regardless of whether Mandelblat-Cerf or Betley is more correct about the natural dynamics of AgRP neurons, our optogenetic protocol is more similar to *either result* than previously described optogenetic protocols.

*6) Figure 5 and Figure 6 are the most interesting and novel, bringing relevant information to the field in apparent contradiction with the experiments described in Betley et al. 2015. However, experiments to characterize the activity of these neurons in the Betley et al. 2015 paper are ignored in this manuscript and not cited. For example, in the Discussion (subsection “AgRP neurons drive feeding through positive reinforcement”), which states that Betley et al. "measured the valence of ongoing AgRP neuron activity," they also measured the valence of silencing the natural activity patterns of AgRP neurons in hunger and found the same results from bidirectional experiments.*

We do not think there is a disagreement between us and the reviewer, and we have rewritten this section to clarify the issue. The full sentence is ‘However this negative reinforcement model was based on experiments that measured the valence of ongoing AgRP neuron activity’. We are not referring here to the difference between activation experiments and inhibition experiments. We are referring to the difference between assigning valence to ongoing AgRP activity (which is what Betley et al. did) and assigning valence to the effects of AgRP neuron firing that persist after the neurons have been silenced (which is what we describe here).

All of the experiments described in Betley et al. were designed to measure the valence of ongoing AgRP neuron activity, regardless of whether they utilized activation or inhibition to make the measurement. That paper does not discuss the pre-stimulation or long-lasting valence effects that are the focus of our paper. We have tried to emphasize this distinction more clearly in the text.

*7) Finally, the direct comparisons to Betley et al. paper should be tempered because although the conclusions are different, so are the experiments employed to make their conclusions.*

The Discussion has been rewritten to emphasize this.